# Phase Shifting Approaches and Multi-Channel Interferograms Position Registration for Simultaneous Phase-Shifting Interferometry: A Review

**Fuzhong Bai** [1,2,*]**, Jiwei Lang** [1]**, Xiaojuan Gao** [3]**, Yang Zhang** [1]**, Jiahai Cai** [1] **and Jianxin Wang** [4]

1    School of Mechanical Engineering, Inner Mongolia University of Technology, Hohhot 010051, China
2    Inner Mongolia Key Laboratory of Special Service Intelligent Robotics, Hohhot 010051, China
3    School of Astronautics, Inner Mongolia University of Technology, Hohhot 010051, China; joan789@163.com
4    Joint Laboratory for Extreme Conditions Matter Properties, Southwest University of Science and Technology, Mianyang 621010, China; wangjianxinkitty@163.com
*    Correspondence: fzbaiim@163.com

**Abstract:** Simultaneous phase-shifting interferometry (SPSI) can simultaneously obtain multiple phase-shifted interferograms and can realize the dynamic wavefront measurement with the use of a phase-shifting algorithm. From the respect of a beam-splitting technique and phase shift achievement of the phase-shifting units, research progress on spatial phase shifting approaches for SPSI systems are classified and summarized, and the key problem affecting SPSI technology is discussed. To ensure the measurement accuracy, it is necessary to perform accurate position registration for multi-channel phase-shifted interferograms before the implementation of a phase-shifting algorithm, and so, the methods of position registration for multi-channel interferograms are also reviewed. This review is expected to prompt research on related fields of phase-shifting interferometry.

**Keywords:** interferometry; simultaneous phase-shifting interferometry (SPSI); phase-shifting approach; multi-channel interferograms; position registration

## 1. Introduction

Phase-shifting interferometry (PSI) has the advantages of high accuracy, full field measurement and non-contact testing. It has been widely used in fields, such as morphology measurement [1], thin film thickness measurements [2], wavefront measurement [3,4], temperature field measurement [5], beam quality evaluation [6], etc. The PSI records multiple phase-shifted interferograms by shifting the reference wave to several different phase values, and then, the quantitative wavefront measurement can be achieved with the phase-shifting algorithm [7,8]. The standard PSI requires a special and constant phase shift, $2\pi/N$, where $N$ is an integer equal to or larger than 3. This requirement is often difficult to meet exactly due to many practical factors, so special efforts must be made to calibrate the phase shifter. To improve the convenience of method, the phase-shifting algorithm with the equal phase shift is further developed to the generalized phase-shifting algorithm dealing with arbitrary phase shifts, for example, least–squares iterative algorithm [9,10], Euclidean matrix norm algorithm [11] and diffraction field statistical averaging method [12] and so on.

Considering the accuracy of phase retrieval, phase-shifting technology is developed from widely used three or four steps to nine or more steps. Considering the requirement of real-time measurement, many advanced two-step PSIs [1,13,14] are also developed. From the spatial-temporal domain of phase shift, the interferometry develops from the temporal phase-shifting technique to the spatial carrier technique and then to the spatial phase-shifting technique.

The temporal PSI needs of $n$ shots to capture all of those phase-shifted interferograms. Thus, it requires the measured phase object and the relevant physical quantities to remain

unchanged as much as possible during the image acquisition process. The temporal PSI is only applicable for static or quasi-static measurement. Besides, it will also extend phase retrieval time.

The spatial carrier method [15,16] is perhaps the simplest technique, which requires the introduction of a known angle between the reference and test waves. That is easy to accomplish, for example, by tilting the reference surface, but it holds two unintended consequences, i.e., the beams are sheared at the CCD detector and the beams often travel different paths through the imaging and/or illumination system. Both of these consequences introduce the systematic error that must be measured or properly calibrated for every case.

The single-shot spatial PSI is another technique to improve the real time of interferometry. It simultaneously captures all needed interferograms with constant phase shift at different spatial positions at the same time and then uses the phase-shifting algorithm to reconstruct the measured wavefront phase. So, it is also known as simultaneous phase-shifting interferometry (SPSI). The primary advantage of the SPSI over temporal PSI is that only single shot is required. The acquisition time is several orders of magnitude smaller than the temporal PSI. Rapid acquisition offers significant vibration immunity and the ability of dynamic measurement.

To build the SPSI systems, one approach is to use multiple cameras and polarization elements to present a different interferogram to each camera. This multi-camera scheme requires simultaneous control of multiple cameras [17]. A more compact approach is called a pixelated mask scheme [3,18,19]. In this approach, the relative phase between reference and test waves is modified on a pixel-by-pixel basis by a micro-polarizer phase-shifting array placed just before the detector. In this manner, a single exposure acquires a multiplexed interferogram. A drawback of this approach is that it is sensitive to polarization aberrations in the optical measurement and difficult to assemble and match with the camera. The third approach is to use a single camera to simultaneously record multi-channel phase-shifted interferograms generated from the combination of beam-splitting components, such as grating, beam splitter, Wollaston prism and polarization optics. The single-camera scheme has the advantage of compact optical structure and low cost, and so, it becomes the most commonly used scheme of simultaneous phase-shifting.

In addition, whether it is a single-camera or a multi-camera scheme, due to the spatial separation of multiple phase-shifted interferograms, position registration and region segmentation for them should be conducted prior to calculating the wavefront phase. Otherwise, a phase retrieval error will be introduced [20–22]. The paper will give a systematic review of typical phase-shifting approaches for SPSI as well as position registration methods of multi-channel phase-shifted interferograms.

## 2. Simultaneous Phase-Shifting Approaches

The key technology of simultaneous phase-shifting is how to split the light beam and accurately introduce phase shift. Most SPSIs adopt the scheme of spatial-splitting and polarization phase-shifting, which uses beam-splitting elements to obtain multiple channels and then introduces different phase shifts in each channel to achieve simultaneous phase-shifting. The beam-splitting elements currently used in SPSI mainly include polarizing or non-polarizing beam splitter, diffraction element such as grating and holographic optical element. But the phase-shifting process is always implemented by wave plates, polarizers or gratings. The structure of SPSI systems is generally complex and not compact enough. To simplify the optical structure, most SPSIs use a three-step or four-step phase-shifting method with the phase step of $\pi/2$.

### 2.1. Beam Splitter-Splitting and Polarization Phase-Shifting

The earliest SPSI was implemented by Smythe et al. [23] using polarization elements, such as half wave plate (HWP), quarter wave plate (QWP) and polarizing beam splitter (PBS), which is shown in Figure 1. Four frames of interferograms with $\pi/2$ phase shift are

generated. To ensure the simultaneity of signal acquisition, this system uses one signal source to simultaneously control four CCD cameras.

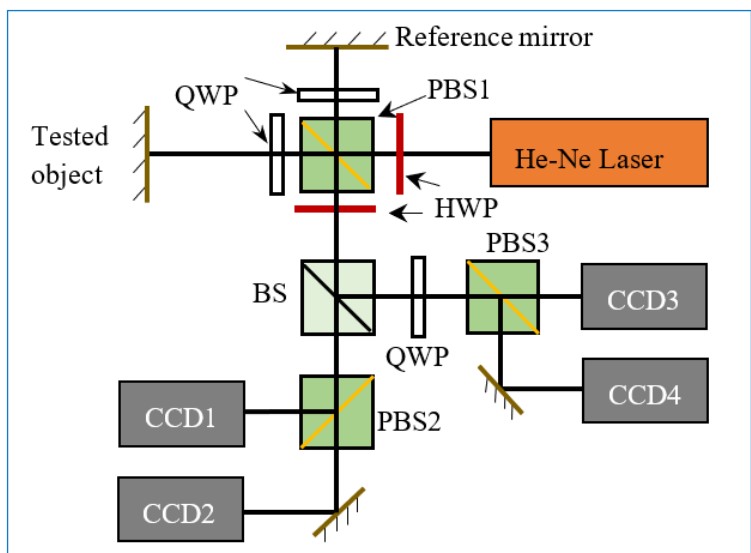

**Figure 1.** Optical schematics of Smythe system.

Szwaykowski et al. [24] proposed a Fizeau-type SPSI based on optical coating technology and polarization phase-shifting technique in 2004. The key component is an amplitude-type beam-splitting prism, as shown in Figure 2. This system is a typical representative of beam-splitting techniques. Reference and test beams with orthogonal linear polarization state occur on the beam-splitting prism along the same path and then are split into four groups. Three groups of them are selected, and different phase shifts are introduced by adding, respectively, appropriate QWP and polarizer (P) to each channel path. Similar to the above devices, there are also the Koliopoulos system [25] and the Haasteren system [26].

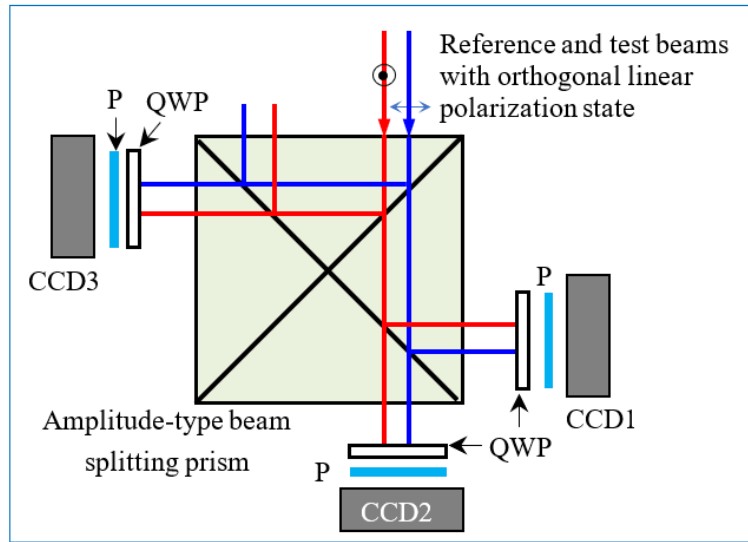

**Figure 2.** Optical schematics of Szwaykowski system.

The multi-camera scheme requires each camera to have completely consistent photo-electric response characteristics. The inconsistency of the surface of optical elements, even dirty spots and uneven coating, can seriously affect the measurement result. In order to meet the consistency requirements, the processing and assembly of all components in the

system become particularly important. The system is very expensive, and its control is more complex.

### 2.2. Polarization Splitting and Polarization Phase-Shifting

Notaras et al. [27,28] designed a polarization simultaneous phase-shifting structure, as shown in Figure 3, which can achieve spatial phase-shifting interference between two incidence reference and test beams with orthogonal linear polarization state. With the use of the quarter wave plate (QWP), beam splitter (BS) and polarizing beam splitter (PBS), four phase-shifted patterns at different spatial positions are imaged onto a CCD camera. This not only ensures synchronous acquisition but also avoids the defects of multiple cameras and reduces costs. This phase-shifting approach has also been applied by us to common path radial-shearing interferometry [29,30] and point diffraction interferometry [31]. Four-channel phase-shifted interferograms with $\pi/2$ phase shift generated from point diffraction interferometry [31] are shown in Figure 3.

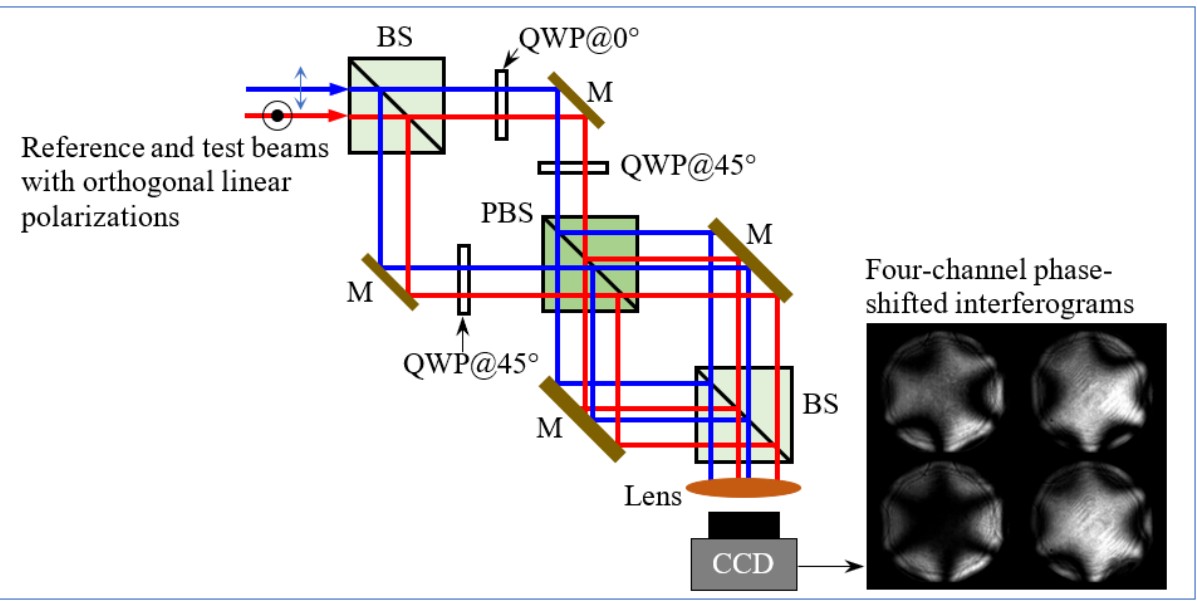

**Figure 3.** Optical schematics of Notaras system and four-channel interferograms.

Toto-Arellano et al. [32] used three coupled interferometers to obtain four-channel optical paths. Then, by placing linear polarizers in each obtained replica, the controllable phase shifts can be achieved.

The polarization splitting and polarization phase-shifting approach allows the device to be free of diffractive or holographic elements and only uses the common components, such as BS and PBS, to implement beam splitting. The disadvantage is that the use of more elements leads to larger structure and more difficult adjustment of the system.

### 2.3. Holographic Element Splitting and Polarization Phase-Shifting

Millerd et al. [33,34] developed a SPSI based on a specially designed two-dimensional orthogonal holographic optical element (HOE) in 2001, and the optical structure diagram is shown in Figure 4. The linearly polarized beam passes through a Twyman-Green system and then is split into reference and test beams with the orthogonal linear polarization state. After passing through a HOE, the test and reference beams are diffracted into four groups; then, they pass through a four-polarizer array mask to form four phase-shifted interferograms with $\pi/2$ phase shift, which are simultaneously imaged on a CCD camera. This system optimizes the optical structure of SPSI, but the HOE used in this system cannot be used for multi-wavelength or white light interferometry, and the mask fabrication is difficult.

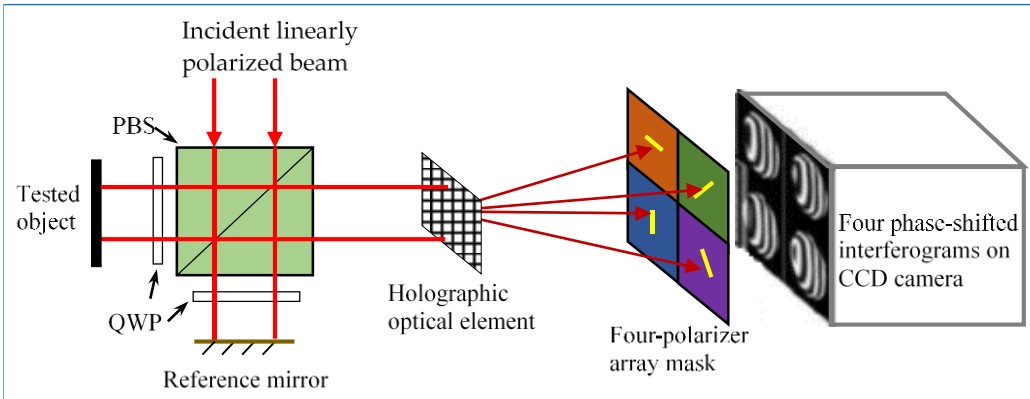

**Figure 4.** Optical schematics of holographic splitting and polarization phase shifting.

Millerd et al. [35,36] further improved the phase mask in 2004 by replacing the original HOE splitting and four-polarizer mask with a micro-polarizer array mask. The concept of pixelated phase-mask dynamic interferometry is shown in Figure 5. A reference beam and a test beam have orthogonal polarization states (which can be linear as well as circular) with respect to each other. A pixelated phase mask introduces an effective phase shift between the reference and test beams at each pixel and subsequently interferes with the transmitted light. A unit cell is comprised of four discrete phase shifting units in which linear polarizers are set at angles of 0, $\pi/4$, $\pi/2$ and $3\pi/4$ for the four desired phase shift values. The phase shifts introduced then correspond to a 0, $\pi/2$, $\pi$ and $3\pi/2$ relative phase between the test and the reference beams. In the phase retrieval, four pixels in a unit cell need to be calculated using four steps of a phase-shifting algorithm. Its technical principle is similar to the polarization phase-shifting mentioned above, which is equivalent to combining four spatially separated phase-shifted interferograms into a single frame interferogram.

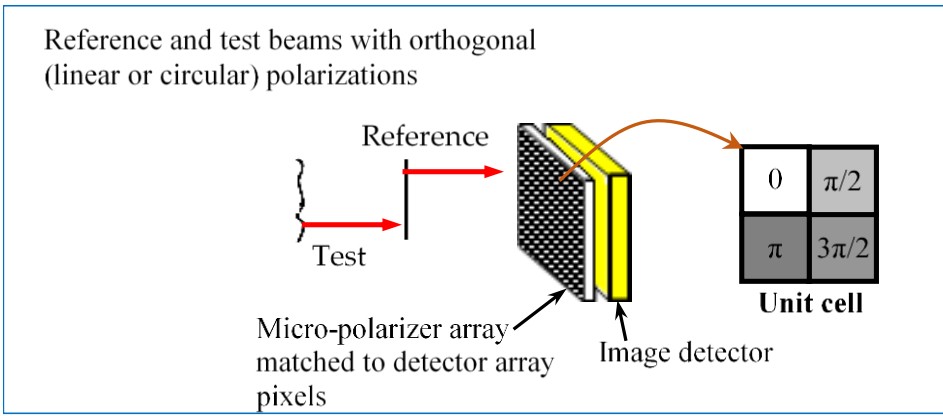

**Figure 5.** Optical schematics of pixelated phase-mask dynamic interferometry.

This pixelated mask can be applied to various polarization interferometry to achieve dynamic measurement, but its disadvantage is that the micro-polarizers array mask needs to be strictly matched to detector array pixels and the mask is difficult to manufacture accurately.

### 2.4. Grating Splitting and Polarization Phase-Shifting

The idea of grating splitting simultaneous phase-shifting technology originated from the spatial PSI based on the combination of grating and pinhole diffraction proposed by Kwon [37]. The diffraction wave through the pinhole in the amplitude type cosine grating is used as the reference wave. The reference wave interferes respectively with the 0 and ±1 order diffracted light and hence obtains synchronously three frames of phase-shifted interferograms. The grating in the system has the functions of splitting and phase shifting.

Hettwer et al. [38] used one-dimensional grating splitting, quarter wave plate (QWP) and polarizer to achieve synchronous phase-shifting, and the optical structure is shown in Figure 6. Two orthogonal linearly polarized beams pass through the grating and then are diffracted. The 0 and ±1 order of diffracted light is selected, and the QWP is inserted into an (+1, −1) order diffracted path, respectively. By adjusting QWP, three frames of phase-shifted interferograms with −π/2, 0 and π/2 phase shifts are generated simultaneously. Due to the influence of grating diffraction and wave plate transmittance, three patterns from this system have different contrast and modulation amplitude, which affects the measurement accuracy of the system. Qian et al. [39] applied simultaneous phase shifting to Mach-Zender interferometry for dynamic measurement of liquid refractive index, in which the first-order diffraction energy from the used 2D Ronchi phase grating reaches about 60% of the total incident light energy.

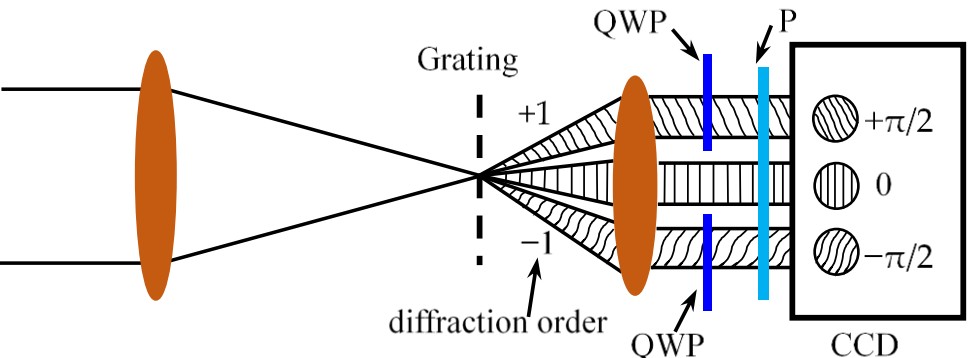

**Figure 6.** Optical schematics of holographic splitting and polarization phase-shifting.

Chen et al. [40–42] have established a SPSI based on 2D diffraction grating splitting and polarization phase-shifting technique, which is shown in Figure 7. In the polarized Twyman-Green structure, the measured information is introduced and two orthogonal linearly polarized beams emit through the PBS. After passing together through a QWP with the fast axis of π/4 with respect to the x-axis, two linearly polarized beams become left-rotation circularly polarized light and right-circularly polarized light, respectively. By rotating the QWP placed in front of the grating, equal amplitudes between reference and test beams can be obtained.

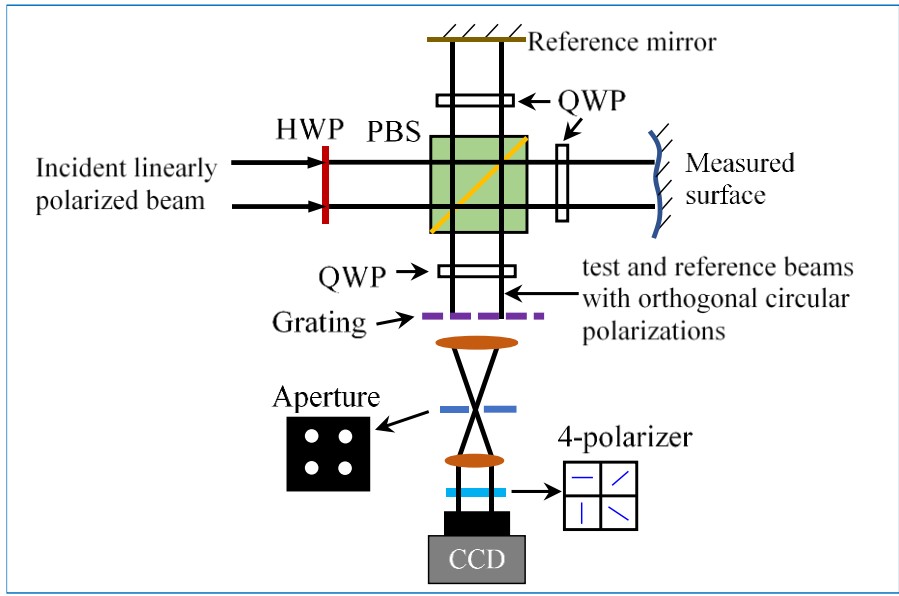

**Figure 7.** Optical schematics of grating splitting and polarization phase-shifting.

On this basis, a 2D orthogonal grating is used for beam-splitting, and ($\pm 1$, $\pm 1$) order of diffracted light is selected as the phase-shifting unit so that the four groups of diffraction light have the same diffraction efficiency. Four groups of diffracted beams irradiate on the polarizer mask to generate four phase-shifted interferograms simultaneously imaged on a CCD camera. The polarizer mask including four linear polarizers is placed at angles of 0, $\pi/4$, $\pi/2$ and $3\pi/4$ in a rectangular arrangement in a clockwise direction.

Similarly, Islas-Islas et al. [43] implemented an experimental system based on a polarized Mach–Zehnder interferometer coupled to a 4f system with a 2D diffraction recycled grating in the Fourier plane. Some patterns showed minimal variations in intensities. For this reason, it was necessary to use a pre-filtering process in order to eliminate the background, normalize the amplitude modulation and filter out noise.

The difficulty of this technology is that 2D grating requires precise processing. It is difficult to ensure that the separated diffraction beams hold a consistent light intensity distribution. So, the uneven distribution of light intensity between several interferograms affects the accuracy of wavefront retrieval.

### 2.5. Key Problems Affecting SPSI Technology

The most attractive advantages of SPSI are single-shot acquisition, dynamic measurement, insensitivity to vibration and high retrieval accuracy. There are still the following key problems in the SPSI technology at present.

(1) Intensity consistency and spatial distribution consistency of multi-channel beam. This is related to the consistent intensity of beam-splitting light and the consistency of optical elements in different phase-shifting channels, as well as the consistency of photoelectric characteristics of multiple detectors or at different positions of one detector. It will affect the background intensity and fringe visibility and has a significant impact on the retrieval accuracy of the phase-shifting algorithm.

(2) Accuracy of phase shifts. The phase step is generally $\pi/2$. The phase shift accuracy directly determines the algorithm accuracy, which is a key factor determining the measurement accuracy in the SPSI [44].

(3) Spatial resolution. The SPSI has gradually developed from the original multi-camera system to the single-camera system using a CCD camera to record multiple interferograms. This requires CCD to have sufficient spatial resolution; otherwise, it will also reduce the accuracy of phase retrieval.

(4) Most SPSI systems adopt polarization phase-shifting interference schemes, and the stress deviation of optical components in the system and the measured phase object will affect the measuring results [45]. The simultaneous phase-shifting approach based on inclination angle deflection into the oblique incidence interferometry is also developed [46,47], but this approach has the limitation in measuring range. New phase-shifting schemes need to be further developed in the future to expand the application field and flexibility of SPSI.

(5) Position registration of multi-channel interferograms. Due to the fact that phase-shifted interferograms are captured at different spatial positions, accurate position registration of the interferograms must be performed before wavefront retrieval so as to ensure the measuring accuracy. Also, this needs in-depth research in designing the SPSI system.

## 3. Multi-Channel Interferograms Position Registration Methods

In addition to three- or four-channel SPSIs, dual-channel PSIs [48–53] are also widely studied. Whether obtained from one or multiple CCD cameras, multi-channel phase-shifted interferograms require spatial position registration for each channel interferogram before phase retrieval. Inaccurate position registration will be converted into phase shift error and will result in measuring errors [1,54,55]. Therefore, spatial position registration of multi-channel phase-shifted interferograms is vital to all SPSIs.

### 3.1. Calibration Plate-Assisted Method

Because there are phase shifts between each channel interferogram, we cannot use conventional image processing technology to directly implement position registration of phase-shifted interferograms.

Chen et al. [56] inserted a target object into the optical path, besides, another means is to place cross-line [57,58], marker board [51] and other marker points. In this case, the image of the target object is included in each channel pattern. Then, by using correlation operation, we can obtain the position-matching relationship, or by using the feature detection algorithm, we can calculate the center of spot contour, the gravity of spot region or the coordinates of marker points to determine the relative position between each channel's interferograms.

Li et al. [51] added a calibration plate in the optical path of a dual-channel PSI. The size of the calibration plate is 0.25 mm × 0.25 mm, including 10 × 10 checks. Two images are captured simultaneously by two different cameras in the interferometer at one shot to the calibration plate. The homonymy corners in two images of calibration plate were established and labeled according to the corner extraction algorithm. The transform matrix between dual-channel images is established, and then, the dual-channel interferograms matching, which is based on the transform matrix, is realized.

Jin et al. [59] used a cross-shaped silicon template as the target object and obtained four-channel background images. By using the speed-up robust features (SURF) algorithm to detect features in the images and by using the random sample consensus (RANSAC) algorithm to calculate the translation and rotation transform relationships between each channel images, the transform matrix is obtained, which is used for the multi-channel phase-shifted interferograms to realize position registration. This scheme is more applicable and effective for multi-camera SPSI systems.

### 3.2. Correlation Algorithm

Kiire et al. [60] obtained extremely low contrast interferograms by adjusting the optical path so that the four patterns had similar gray distribution, and they used the spatial cross-correlation algorithm to determine the position matching relationship between the interferograms. Zheng et al. [22,61] used the phase correlation algorithm (PCA) for the test light spot image to obtain position registration. Its principle is similar to that of Kiire's algorithm [43]. The only difference is that Kiire's algorithm is a spatial domain correlation algorithm and Zheng's algorithm [61] is a frequency domain correlation algorithm. The theoretical basis of PCA is based on the translation characteristic of 2D image Fourier transform, which means that the displacement in the spatial domain is equivalent to the phase shift in the frequency domain.

On the basis of the 2D grating splitting SPSI system as shown in Figure 7, Zheng et al. [22,61] first obtained the simultaneous phase-shifted interferograms and then rotated the half wave plate (HWP) to eliminate the reference light, thereby obtaining four test light spots. The processing flowchart of the PCA is shown in Figure 8. The detailed steps of the algorithm are described as follows.

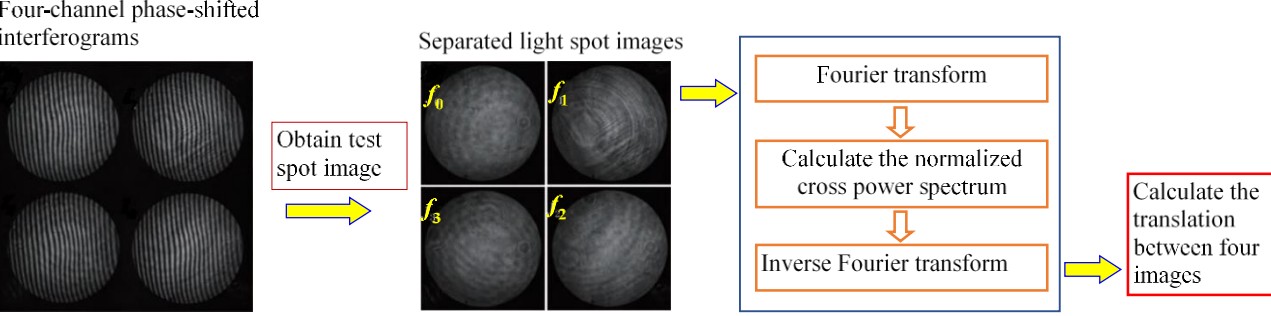

**Figure 8.** Processing flowchart of phase correlation algorithm (PCA).

(1) Four test light spots are firstly separated and are seen as four independent images.

(2) One spot is defined as the reference spot.

(3) Perform Fourier transform on the reference spot image ($f_0$) and one of the test spot images ($f_1$), respectively.

(4) Calculate their normalized cross power spectrum and perform inverse Fourier transform to obtain the unit pulse function:

$$C(x,y) = \mathcal{F}^{-1}\left(\frac{\mathcal{F}\{f_1\}\cdot\mathcal{F}\{f_0\}^*}{|\mathcal{F}\{f_1\}|\cdot|\mathcal{F}\{f_0\}|}\right) = \delta(x - \Delta x_1, y - \Delta y_1) \qquad (1)$$

where $(\Delta x_1, \Delta y_1)$ is the translation amount of $f_1$ relative to $f_0$ along the $x$-axis and $y$-axis. The asterisk "*" represents a conjugate operation, and $\delta(\cdot)$ is the Dirac function.

(5) Calculate the coordinates of the peak point so as to obtain the translation $(\Delta x_1, \Delta y_1)$ between the two spot images.

(6) Similarly, obtain the position relationship between the other spot images and the reference spot image.

### 3.3. Variance Analysis Method Based on Sequence Interferograms

We established a self-referencing point diffraction SPSI [31], which uses a CCD camera to record four-channel phase-shifted interferograms with the phase step of $\pi/2$. The developed variance analysis method for sequential interferograms introduced statistical analysis, region segmentation and the improved Hough transform to realize position registration of circular aperture multi-channel phase-shifted interferograms [21]. The processing flowchart of the algorithm is shown in Figure 9 and included as follows.

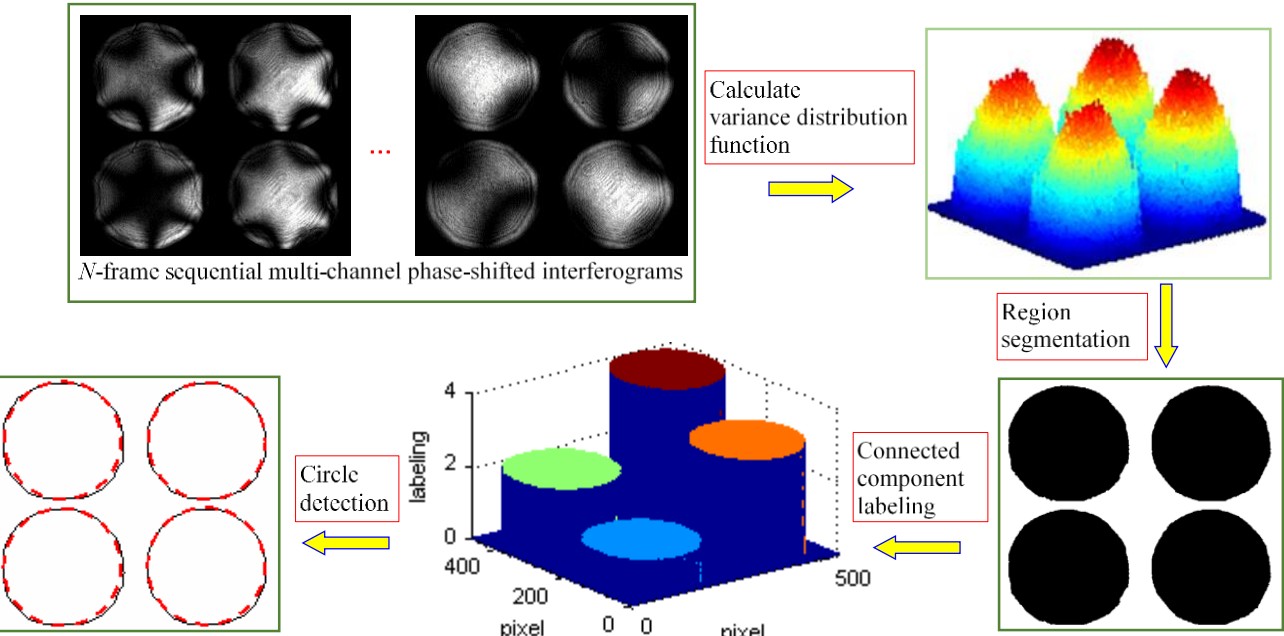

**Figure 9.** Processing flowchart of the variance analysis algorithm.

(1) Introduce random phase modulation between two interference beams in the built SPSI and record $N$-frame sequence multi-channel phase-shifted interferograms. In this case, the intensity of sequence patterns can be expressed as

$$I_k(x,y) = A(r)\{a + b\cos[\varphi_k(x,y) + \delta]\} \qquad (2)$$

where $a$ and $b$ are the background and modulated intensity of interferogram, respectively; $\varphi_k$ ($k = 1, \ldots, N$) is the introduced phase difference; and $\delta$ describes the phase

shift. The experiments confirm that the proposed algorithm is effective when $N > 10$. $A(r)$ is the circular aperture function, which can be expressed as

$$A(r) = \begin{cases} 1, \ r \leq r_0 \\ 0, \ r > r_0 \end{cases} \tag{3}$$

where $r$ is radius of circular aperture.

(2)    Calculate the variance value at a certain point $(x_0, y_0)$ in $N$-frame patterns, i.e.,

$$D(x_0, y_0) = \frac{1}{N} \sum_{k=1}^{N} \left[ I_k(x_0, y_0) - \overline{I}(x_0, y_0) \right]^2 \tag{4}$$

where $\overline{I}(x_0, y_0)$ is the averaging intensity of the point $(x_0, y_0)$ in $N$-frame patterns. The 2D variance distribution function $D(x, y)$ is obtained after all pixels in the pattern undergoing the above operation.

(3)    Apply the Otsu's thresholding algorithm to $D(x, y)$ to segment the background and interference regions.

(4)    Perform connected component labeling and Hough transform circle detection so as to obtain the circular contour parameters of each object region.

(5)    According to the detected parameters, extract effective data of each channel. The measured phase is obtained with the four-step phase shifting algorithm as follows:

$$\delta(x, y) = \tan^{-1} \left( \frac{I_{3\pi/2}(x, y) - I_{\pi/2}(x, y)}{I_0(x, y) - I_\pi(x, y)} \right) \tag{5}$$

Figure 10 shows the position registration result and reconstructed phase of an experimental four-channel phase-shifted interferogram.

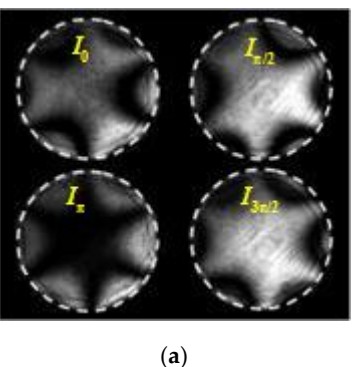 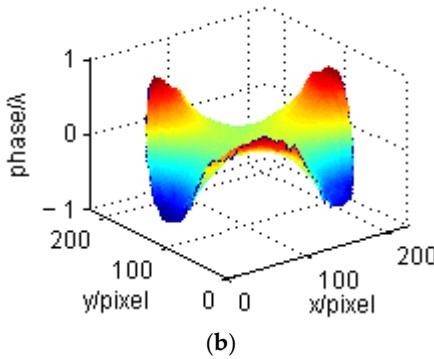

(**a**)                         (**b**)

**Figure 10.** Position registration result (**a**) for an experimental four-channel phase-shifted interferogram and the wavefront phase (**b**) reconstructed from the experimental pattern.

## 4. Discussion

Compared with the temporal PSI, such as four-step phase-shifting method, the SPSI can measure the amplitude and wavefront of the incident light beam in real time and needs no stable and regular intensity distribution. It is very important for the wavefronts measurement of laser beam whose amplitude and phase distribution are very irregular and unstable.

In the SPSI, the simultaneous phase-shifting approach is an important and key technology. At the same time, the simultaneous phase-shifting approach is also a key technology in the dynamic digital holography with a large field of view and a high resolution [19,62,63], and dynamic imaging the three-dimensional topography of structures, particularly in the case of translucent or scattering samples [27], such as imaging of flow and sound [64].

For the four typical simultaneous phase-shifting approaches included in the paper, their difference mainly lies in the difference of beam splitting methods, which includes beam splitter splitting, polarization splitting, holographic element splitting and grating

splitting. The phase shifts are always introduced by the polarization elements. Their advantages and limits include the following.

(1) The beam splitter splitting SPSI belongs to multi-camera spatial phase-shifting system. It is difficult to ensure the consistency in the photoelectric response and exposure time of multiple cameras, so that will introduce phase shift errors and affect measuring accuracy. The other three SPSIs belong to a single-camera system, which can avoid the above problem but will result in low spatial resolution.

(2) The polarization splitting SPSI uses more optical elements and so leads to incompact structure and difficult adjustment of system.

(3) The HOE splitting SPSI optimizes the optical structure of system, but the mask fabrication is difficult. The mask with micro-polarizer array needs to be strictly matched to array pixels of CCD camera.

(4) For the grating splitting SPSI, the separated diffraction beams from 2D grating may hold inconsistent light intensity distribution. That will result in a decrease in the accuracy of wavefront retrieval. There are still some urgent problems to be solved in the development of SPSI to this day. The work developing a compact, easy adjustment and accurate phase shift SPSI is an important research direction.

The output results of SPSI are multi-channel phase-shifted interferograms from different spatial positions, so the accurate position registration of multi-channel interferograms must be performed before wavefront retrieval. The three typical position registration methods are discussed as follows from the aspects of convenience, application performance and mismatch error.

(1) Convenience analysis of algorithms. Most presented algorithms belong to indirect calculation methods, which include the calibration plate-assisted method and the correlation algorithm. It needs to place target objects in the optical path; the accuracy of target feature extraction affects the accuracy of position registration; and the repeatability of registration results is poor. The PCA needs to suppress the reference wave and uses the position of test spots to indirectly represent that of multi-channel phase-shifted interferograms. Because the interference region may not completely match the test spot, the calculated deviation will be impossible to avoid. For some interferometer systems, such as common path structure and non-polarizing interference, it is very difficult to obtain test spot image. For the variance analysis method, the multi-channel interferograms obtained in experiment are directly processed. It has the advantage of intuitive result, simple principle and convenient operation.

(2) Application performance of algorithms. The common problem for most algorithms is that, even if position registration is performed, the interference region cannot be obtained simultaneously. Further interference region segmentation is necessary. In fact, the extraction of effective region in interferograms is also very important for accurate phase retrieval and phase unwrapping [65–67]. For example, for the PCA applied to the case of multi-channel interferograms from one camera, it is necessary to firstly segment the regions of each channel so as to perform registration operation, which limits application flexibility. The variance analysis method cannot only perform registration but can also extract the interference region of each channel image. It has more wide application and better universality.

(3) Mismatch error compensation. Servin et al. [68,69] pointed out that the mismatch errors of interferogram can affect subsequent phase retrieval and proposed an improved phase demodulation algorithm to reduce the effect of mismatch errors. This current algorithm is only applicable to pixelated interferogram. In addition, Kimbrough et al. [70] used higher-order equations to represent respectively the constant term and interference term of fringe patterns and further reduced the effect of the constant term on the retrieval result through optimizing the higher-order algorithm. This is also the algorithm for reducing mismatch error.

## 5. Conclusions

Throughout the development of PSIs in the past few decades, the research work mainly focused on high-speed, high spatial resolution, high accuracy, large field of view, noise resistance, vibration resistance and low cost. Many achievements are made in these fields. Especially in order to achieve real-time dynamic measurement, the PSI has developed from the temporal PSI to the SPSI. This paper reviews four typical phase-shifting techniques used in the SPSI, which include optical principle, technical development, and advantages and limits.

In addition, the spatial position registration of multi-channel phase-shifted interferograms is also vital to all SPSIs. This paper further reviews three typical position registration methods, and their convenience and application performance are also discussed. There are the phase shifts between multi-channel interferograms. The presence of diffraction fringes and the extension of interference fringes at the boundary will lead to blurry contours of interferograms. These factors increase the difficulty of region extraction and position registration. Also, the mismatch error will cause the measuring error, so the position registration should be considered in the design of the SPSI system. The technology that can determine the position relationship of each channel pattern and extract interference regions at the same time is more effective and has better practical value. In summary, the position registration of multi-channel interferograms cannot be solved through the simple optimization of algorithm. Further research on a universal and highly automated position registration algorithm is necessary, and position registration based on interference region segmentation is also a more feasible technical solution.

**Author Contributions:** Conceptualization and methodology, F.B.; investigation, J.L. and Y.Z.; writing—original draft preparation, F.B.; writing—review and editing, X.G., J.C. and J.W.; funding acquisition, F.B. and X.G. All authors have read and agreed to the published version of the manuscript.

**Funding:** This research was funded by the National Natural Science Foundation of China under Grants 62165011 and 51765054, the Natural Science Foundation of Inner Mongolia of China under Grant 2022MS0601 and the Science and Technology Plan Projects of Inner Mongolia of China under Grant 2021GG0263.

**Institutional Review Board Statement:** Not applicable.

**Informed Consent Statement:** Not applicable.

**Data Availability Statement:** The data presented in this study are available on request from the corresponding author.

**Conflicts of Interest:** The authors declare no conflict of interest.

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
