# Peer review of "Phase Shifting Approaches and Multi-Channel Interferograms Position Registration for Simultaneous Phase-Shifting Interferometry: A Review"

_photonics, doi:10.3390/photonics10080946_

Round 1

Reviewer 1 Report

This manuscript reviews the phase shifting approaches and multi-channel interferograms registration techniques currently used in SPSI. The manuscript is well organized, and extensively describes the simultaneous phase shift methods based on different beam splitting techniques, as well as the interferograms position registration methods based on calibration plates, correction algorithms and variance analyzation.

I think this is a relatively comprehensive and valuable review. As my comments are addressed, I would recommend that this manuscript can be accepted.

1. A large number of images in the manuscript are blurry, please revise them.

2. The accuracy and effectiveness of different position registration methods should be introduced and compared quantitatively.

Author Response

Thanks for the review and suggestions of reviewer.

1) A large number of images in the manuscript are blurry, please revise them.

Response:  All images in the manuscript have been edited again.

2) The accuracy and effectiveness of different position registration methods should be introduced and compared quantitatively.

Response:  This paper reviews three typical position registration methods, and their convenience and application performance are also discussed in the Section 3.4.  The calibration plate assisted method (method #1) needs an additional target object inserted into the optical path. The correlation algorithm (method #2) needs to obtain the test light spot images after suppressing reference light.  The variance analysis method (method #3 ) needs to obtain multiple frames of  sequential interferograms. The image sources of three methods are different. At the same time, accurate contours and theoretical centers of multi-channel interferograms cannot been obtained in the experiment. Therefore, we only analyze the technical ideas, algorithm principle, advantage and disadvantage of methods, and technical difficulties of position registration between phase-shifted patterns. Nevertheless, an integrated outlook of the mentioned methods  is supplemented in the conclusion section. The modified part is displayed in red text in the revision manuscript.

Reviewer 2 Report

This manuscript reviews the research progress on phase shifting approaches and multi-channel interferograms position registration for SPSI systems. Overall, the manuscript is written well. I think this paper would be acceptable after commenting the following comments.

 An integrated outlook of the four typical phase-shifting technology should be summarized in Conclusion.

The manuscript needs careful editing and particular attention to English grammar, spelling, and sentence structure, for example:

---In Abstract, Line 16, “classified summarized “would be “classified and summarized “

---Line 134, “Toto-Arellano et al. used three coupled interferometers obtain four-channel optical paths….” would be “Toto-Arellano et al. used three coupled interferometers to obtain four-channel optical paths…..”

---Line 221, “…makes the accurate wavefront retrieval become difficult…” would be “…makes the accurate wavefront retrieval difficult…”

---Line 389, “…and many achievements have been made in these fields.” would be “…and many achievements being made in these fields.”

Author Response

Thanks for the review and suggestion of reviewer.

1) An integrated outlook of the four typical phase-shifting technology should be summarized in Conclusion. 

Response: The relevant description has been modified and added to the Conclusion. 

2) The manuscript needs careful editing and particular attention to English grammar, spelling, and sentence structure. 

Response:  We carefully checks and revises the text and grammar. The modified part is displayed in red text in the revision manuscript.

Reviewer 3 Report

This manuscript reviews multiple approaches to simultaneous phase-shifting interferometry, with a particular focus on experimental setups that use a single sensor to illustrate the relevance and importance of a position registration method proposed by the authors. I believe that this research topic does not align completely with the Aim and Scope of MDPI Photonics; I would suggest transferring it to MDPI Optics, but not without addressing the following concerns that prevent me from recommending the acceptance of this initial revision:

1) The document lacks a consistent narrative and exhibits minor but frequent grammatical errors (particularly within the Introduction section).

2) It is objectively wrong to claim that the PSI technology was first proposed by Yamaguchi et al. in 1997. Some notable counter-examples are Bruning, J.H., Herriott, D.R., Gallagher, J.E., Rosenfeld, D., White, A.D., & Brangaccio, D.J. (1974). "Digital wavefront measuring interferometer for testing optical surfaces and lenses." Applied optics, 13 11, 2693-703 ... Morgan, C. J. (1982). "Least-squares estimation in phase-measurement interferometry." Optics Letters, 7(8), 368-370 ... Greivenkamp, J. E. (1984). "Generalized data reduction for heterodyne interferometry." Optical Engineering, 23(4), 350-352 ... Cheng, Y.Y., & Wyant, J.C. (1985). "Phase shifter calibration in phase-shifting interferometry." Applied optics, 24 18, 3049 ...and many more.

3) Some figures seem to be used without giving credit to the original authors (Figure 5 at the very least; could be others).

4) The manuscript lacks novelty as the main concepts that the authors want to focus on have already been previously published.

Moderate editing of English language is required, mainly due to missing articles (a, an, the, ...) and normal grammar errors for non-native speakers.

Author Response

Thanks for the review and of reviewer. 

1) The document lacks a consistent narrative and exhibits minor but frequent grammatical errors (particularly within the Introduction section). 

Response: We carefully checks and revises the text and grammar. The modified part is displayed in red text in the revision manuscript.

2) It is objectively wrong to claim that the PSI technology was first proposed by Yamaguchi et al. in 1997. 

Response: This description in the original manuscript is inaccurate. These contents are rewritten in the first paragraph of the introduction. 

3) Some figures seem to be used without giving credit to the original authors (Figure 5 at the very least; could be others).

Response: According to the comment of reviewer, we pay special attention to this issue and describe the source of all images in the main text.

For example, for Figure 5,the description in the text is as follows: “Millerd et al. [31, 32] further improved the phase mask in 2004 through replacing the original HOE splitting and four polarizers mask technique with a micro-polarizer array mask, and the concept of pixelated phase-mask dynamic interferometry is shown in Figure 5 ”.

4) The manuscript lacks novelty as the main concepts that the authors want to focus on have already been previously published.

Response:  The simultaneous phase-shifting interferometry (SPSI) is the present popular PSI technology. The paper focus on two important technical problems (phase-shifting approaches and position registration) of SPSI, and systematically summarizes relevant technical achievements, which have certain reference value for the improvement of SPSI performance.

Round 2

Reviewer 3 Report

The core concepts were already published by the same authors in recent publications. The "review" tag is not convincing enough to justify an article that is so derivative of their previous work.

Author Response

This manuscript focuses on the phase-shifting approaches and position registration of the SPSI, and systematically summarizes relevant technical achievements and developement. The core content has not been published.